# Endoscopic Biliary Drainage in Surgically Altered Anatomy

**DOI:** 10.3390/diagnostics13243623

**Published:** 2023-12-08

**Authors:** Marco Spadaccini, Carmelo Marco Giacchetto, Matteo Fiacca, Matteo Colombo, Marta Andreozzi, Silvia Carrara, Roberta Maselli, Fabio Saccà, Alessandro De Marco, Gianluca Franchellucci, Kareem Khalaf, Glenn Koleth, Cesare Hassan, Andrea Anderloni, Alessandro Repici, Alessandro Fugazza

**Affiliations:** 1Division of Gastroenterology and Digestive Endoscopy, Humanitas Research Hospital-IRCCS, Via Manzoni 56, Rozzano, 20089 Milan, Italy; marco.spadaccini@humanitas.it (M.S.); carmelo.giacchetto@humanitas.it (C.M.G.); matteo.fiacca@humanitas.it (M.F.); matteo.colombo@humanitas.it (M.C.); marta.andreozzi@humanitas.it (M.A.); silvia.carrara@humanitas.it (S.C.); roberta.maselli@hunimed.eu (R.M.); fabio.sacca@humanitas.it (F.S.); alessandro.demarco@humanitas.it (A.D.M.); gianluca.franchellucci@humanitas.it (G.F.); cesare.hassan@hunimed.eu (C.H.);; 2Department of Biomedical Sciences, Humanitas University, Via Rita Levi Montalcini 4, Pieve Emanuele, 20090 Milan, Italy; 3Department of Gastroenterology, Saint Michael’s Hospital, University of Toronto, Toronto, ON M5B 1W8, Canada; kkhalaf@hotmail.it; 4Hospital Sultan Ismail, Malaysian Ministry of Health, Johor Bahru 81100, Malaysia; glennkoleth@gmail.com; 5Gastroenterology and Digestive Endoscopy Unit, Fondazione IRCCS Policlinico San Matteo, 27100 Pavia, Italy; a.anderloni@smatteo.pv.it

**Keywords:** EUS, surgery, ERCP, biliary drainage, EUS-BD, altered anatomy

## Abstract

Endoscopic retrograde cholangiopancreatography (ERCP) is considered the preferred method for managing biliary obstructions. However, the prevalence of surgically modified anatomies often poses challenges, making the standard side-viewing duodenoscope unable to reach the papilla in most cases. The increasing instances of surgically altered anatomies (SAAs) result from higher rates of bariatric procedures and surgical interventions for pancreatic malignancies. Conventional ERCP with a side-viewing endoscope remains effective when there is continuity between the stomach and duodenum. Nonetheless, percutaneous transhepatic biliary drainage (PTBD) or surgery has historically been used as an alternative for biliary drainage in malignant or benign conditions. The evolving landscape has seen various endoscopic approaches tailored to anatomical variations. Innovative methodologies such as cap-assisted forward-viewing endoscopy and enteroscopy have enabled the performance of ERCP. Despite their utilization, procedural complexities, prolonged durations, and accessibility challenges have emerged. As a result, there is a growing interest in novel enteroscopy and endoscopic ultrasound (EUS) techniques to ensure the overall success of endoscopic biliary drainage. Notably, EUS has revolutionized this domain, particularly through several techniques detailed in the review. The rendezvous approach has been pivotal in this field. The antegrade approach, involving biliary tree puncturing, allows for the validation and treatment of strictures in an antegrade fashion. The EUS-transmural approach involves connecting a tract of the biliary system with the GI tract lumen. Moreover, the EUS-directed transgastric ERCP (EDGE) procedure, combining EUS and ERCP, presents a promising solution after gastric bypass. These advancements hold promise for expanding the horizons of comprehensive and successful biliary drainage interventions, laying the groundwork for further advancements in endoscopic procedures.

## 1. Introduction and Anatomical Considerations

The need for biliary drainage in patients with surgically altered anatomy (SAA) [1,2] is a more and more frequent challenge for endoscopists given the increasing demand for bariatric surgery and expanding surgical indications for bilio-pancreatic and gastric malignancy [3,4].

Furthermore, biliary obstruction may result from the development of benign or malignant complications of these procedures. As a matter of fact, a complete resection of the gastric lumen and vagal nerve resection, as well as sudden weight loss following bariatric surgery promoted gallstone formation within first years of follow up [5,6]. On the other hand, the Whipple procedure performed for pancreaticobiliary malignancy, already carrying a meager curative rate, may warrant recurrent endoscopic intervening for biliary decompression [7].

While some SAAs may be more prone to develop biliary obstruction, understanding the encountered variation is the first step to deciding the most appropriate approach to endoscopic biliary drainage.

Endoscopic retrograde colangio-pancreatography (ERCP) with the use of side-viewing duodenoscope is the mainstay of treatment in the case of biliary obstruction in normal anatomy conditions, but its technical and clinical success drops significantly in surgically altered anatomies with higher rates of adverse events.

To guide the endoscopic approach to biliary drainage, we can divide SAA in two different categories [8]:-Type I includes conditions in which the duodenum is in continuity with gastric remnant as in the case of sleeve gastrectomy and Billroth I;-Type II includes all the cases in which the stomach is absent (with an esophageal-jejunal anastomosis) or its remnant is not in continuity with the duodenum. This condition is present in Billroth II with gastrojejunostomy, Roux-en-Y gastric bypass (RYGB), Roux-en-Y hepatico-jejunostomy and Whipple’s procedure. Figure 1 summarizes the main surgical interventions and the anatomical alterations.

### 1.1. Type I

-Sleeve gastrectomy

Sleeve gastrectomy is a bariatric procedure that involves the resection of the greater curvature of the stomach, keeping the remnant stomach in continuity with the small bowel.

-Billroth I gastrectomy

This process begins with an antrectomy, followed by creating an end-to-end connection between the remaining stomach and the duodenum. Since the duodenum is continuous with the remaining stomach, ERCP can be performed using the traditional approach.

### 1.2. Type II

-Billroth II partial gastrectomy and gastrojejunostomy

This procedure is nowadays performed in cases of gastric cancer or for complications of peptic ulcer diseases like gastric outlet obstruction (even with gastrojejunostomy alone). An afferent limb of variable length is in continuity with the duodenum and an efferent limb is connected to the jejunum.

-Roux-en-Y gastric bypass

Gastric bypass is a type of bariatric surgery that involves malabsorption. It entails the division of the stomach into a small proximal pouch and a large distal pouch, which remains connected to the duodenum.

The duodenum and proximal jejunum will form the biliopancreatic limb, while a small gastric pouch anastomosed to the distal jejunum will form the Roux limb.

The “mini gastric bypass” is a single-loop procedure that does not need the creation of a Roux’s limb with a single anastomosis between the small gastric pouch and the small bowel. The result will be the presence of an afferent (bilioenteric) and an efferent limb.

-Pancreaticoduodenectomy (Whipple procedure)

The procedure consists of the removal of the pancreatic head, the distal stomach and the duodenum together with the proximal jejunum, distal common bile duct and the gallbladder. In pylorus sparing pancreatoduodenectomy, the distal stomach is preserved. At first, pancreatic duct anastomosis is performed, followed by end-to-side hepaticojejunostomy and eventually gastrojejunostomy or duodenojejunostomy in cases of pylorus preserving procedure.

In cases of type I anatomies, a duodenoscope can be used to reach the Ampulla of Vater and ERCP can be performed using standard accessories. In the case of a failed ERCP procedure, an alternative approach involves endoscopic ultrasound (EUS)-guided access to the bile duct, which can be achieved either through the duodenum or the residual stomach [9,10].

In type II anatomies conventional ERCP is generally not feasible, so different techniques and tools have subsequently been developed over the years [11].

## 2. Forward-View ERCP in SAAs

In conditions of altered anatomy, the use of conventional duodenoscope is roughly unaffected in type I surgeries [12,13,14], but has not been satisfactory in type II given the difficulty of maneuvering the duodenoscope down the Roux limb and in the biliopancreatic limb (considering a short limb 80–100 cm and a long limb 100–150 cm) to reach and gain access to the papilla.

Studies and reports have registered acceptable (62.5–86.1%) rates in approaching the papilla after Billroth II gastrectomies, but the percentages are very low in cases of more complex surgeries, such as RYGB (75.3%) or Whipple’s procedure (57.9%) [13,15].

The use of a pediatric colonoscope for diagnostic and therapeutic ERCP in long-limb bypass patients with a Roux-en-Y anatomy was first described in 1988 [16] and later in 1998. Elton et al. described their experience in 18 patients. In the latter study, when the procedure with the colonoscope was unsuccessful, it was reattempted with an enteroscope, demonstrating an overall success rate of 84% and cannulation rate of 94% [3].

Our group recently proposed the use of the underwater technique (u-ERCP) using a cap-assisted pediatric colonoscope in six patients with altered anatomy, achieving a success rate of 100% without any adverse events (AEs). The underwater technique indeed helps stabilize and straighten the intestinal loops, while the distal attachment cap helps flatten folds, giving stability and improving visualization with an easier cannulation of the papilla (Figure 2) [17] (Table 1).

More recently, Lee et al. conducted a retrospective study including 47 patients to evaluate the feasibility of using a cap-assisted regular colonoscope as the primary approach for ERCP in Roux-en-Y reconstruction.

They also compared the type of jejuno-jejunal anastomosis in these patients showing a higher intubation success rate using the cap-assisted colonoscope in the side-to-side jejunojejunostomy group than that in the side-to-end jejunojejunostomy group (34 of 38 (89.5%) vs. 1 of 9 (11.1%), *p* < 0.001). The presence of a side-to-side jejunojejunostomy was a predictive factor for successful intubation with this technique [24].

Device-assisted endoscopy (DAE) has different devices and techniques developed over the years and can nowadays be performed with single-balloon enteroscope (SBE), double-balloon enteroscope (DBE) or spiral enteroscope (SE).

A long (200 cm) and short (155 cm) length of DBE and SBE are now available, with a working channel of 2.8 mm or 3.2 mm.

In addition to the lack of side-view orientation and an elevator, another limitation of this procedure is the length of the endoscopes and the dimension of the operative channel (2.8 mm) that sometimes preclude the use of conventional stents and accessories [3].

The introduction of short-length, double-balloon (sDBE) and short-length, single-ballon (sSBE) enteroscopes (155 cm of length, with a working channel of 3.2 mm) helped overcome these disadvantages.

In 2006, sDBE was first used to perform ERCP in an RYGB patient.

DBE was shown to reach the papilla or the anastomosis in 89% of cases in a systematic review. In those cases, a cannulation success rate of 93% and a therapeutic success rate of 82% were reported. With SBE, reaching the papilla or anastomosis was possible in 82% of cases, with a cannulation success rate of 86% and an overall therapeutic success rate of 68%.

It was also underlined that the success rate mostly depended on the length of the limb, with the lowest rate registered in RYGB, followed by pancreaticoduodenectomy and Roux-en-Y hepaticojejunostomy, with the best results in Billroth II patients, demonstrating equivalent cannulation rates in patients with both native papilla and biliary-enteric or pancreatico-enteric anastomoses [4]. Overall, with the use of sDBE-ERCP, a success rate between 70.7 and 96% has been described, while SBE-ERCP appeared to be as effective as sDBE, with a success rate between 73 and 92.3% but with shorter insertion and overall procedure time [18,25,26,27].

SE has been introduced in an attempt to overcome the drawbacks of standard enteroscopy procedures. In two studies, Ali et al. and El Zouhairi et al. demonstrated success among RYGB patients, reaching the papilla in 86% and 76.2% of cases, respectively. Once the papilla was reached, the cannulation and therapeutic intervention were successful in 92.3 and 100%, with overall success rates for SE-ERCP being 86% and 64.3%, respectively [19,28] (Table 1).

## 3. EUS-Guided Biliary Drainage Procedures

Biliary access in the case of altered anatomy is one of the most evident paradigms of the complementarity between EUS and ERCP [29]. In this context, dedicated goal-based consent forms adequately informing patients about the complete endoscopist armamentarium should always be available [30,31]. EUS-guided biliary drainage can be executed employing three approaches: EUS-rendezvous (EUS-RV), transmural and EUS-guided antegrade interventions (EUS-AI). In EUS-RV, biliary ducts can be reached with a needle either from the stomach or the duodenum. Then, a guidewire is passed through the dilated biliary system toward the papilla and within the duodenum, where it is captured using either a duodenoscope or an enteroscope after careful exchange of endoscopes to prevent guidewire slippage. The EUS-transmural approach involves the formation of a connection between a segment of the biliary system and the gastrointestinal tract lumen. This can be achieved between the bile duct and the duodenum as in EUS- choledochoduodenostomy (EUS-CDS), or intra-hepatic ducts and the stomach as in EUS-hepaticogastrostomy (EUS-HGS).

EUS-AI consists of access to the biliary tree either from the stomach or the duodenum. A guidewire is negotiated across the stricture through the papilla in an antegrade fashion with the successive positioning of a stent to obtain biliary drainage.

In the case of failed conventional ERCP, an EUS-guided approach can be an efficient alternative to colonoscopy or DAE-ERCP, and the technique of choice depends on the different anatomy alterations. It should be noted that the procedures should be performed at a tertiary center with significant expertise.

In type I anatomies, the papilla is reachable with standard instruments, and this permits standard side-view ERCP, rendezvous, antegrade and transmural procedures like EUS-CDS as in native anatomy, in the case of ERCP failure.

EUS-RV might sometimes be feasible in type II cases. However, when the papilla is not reachable, antegrade approaches by puncturing the bile duct or transmural approaches like EUS-HGS or EUS-hepatico-jejunostomy (EUS-HJ) need to be performed. The focus of this review is on these situations.

The vast majority of the current literature on this topic consists of retrospective case series or non-comparative cohort studies, while randomized controlled trials are absent on this issue. “In a recent meta-analysis [32] investigating outcomes of EUS-biliary drainage (EUS-BD) procedures, technical success of 97.8%, and clinical success of 94.9%, with adverse event rates of 12.8% were reported. It is worth mentioning, though, that this study revealed a wide range in the prevalence of SAA, varying from 14.5% to 100%, which indicates a significant level of data heterogeneity”.

## 4. EUS-Guided Antegrade Intervention

Iwashita and colleagues [20] conducted a prospective study demonstrating the feasibility and safety of EUS-guided antegrade biliary stenting as a treatment for unresectable malignant biliary obstruction (MBO) (Figure 3). In their research, both the technical and clinical success rates for EUS-AI were 95%. In one instance, the procedure was unsuccessful because the left lobe of the liver could not be visualized via EUS and required percutaneous biliary drainage. Adverse events, including mild pancreatitis in three patients and mild fever in one patient, occurred in 20% of the cases (4 out of 20), all of which were effectively managed using conservative methods (Table 1).

There is a lack of sufficient evidence comparing EUS-AI to alternative procedures such as DAE-ERCP or PTBD. In another study conducted by Iwashita and colleagues [21], a total of 64 patients were included, with 35 undergoing EUS-AI and 29 treated by PTBD. The technical and clinical success rates for the EUS-AI and PTBD groups were 97.1% vs. 96.6% (*p* = 1.00) and 97.1% vs. 93.1% (*p* = 0.586), respectively. The rate of adverse events was 11.4% vs. 27.6% (*p* = 0.119). This study concluded that both EUS-AI and PTBD were effective drainage methods, although PTBD exhibited a tendency toward a higher adverse event rate, largely due to events related to the drainage tube.

Recently, the effectiveness of combining EUS-AI with a self-expandable metal stent (SEMS) and EUS-HGS (or EUS-HJS) has been investigated. Ogura and colleagues [33] conducted a multicenter prospective pilot study assessing EUS-HJS combined with antegrade stenting for MBO, and reported a technical success rate of 85.7% (40 out of 49) and an adverse event rate of 10.2% (5 out of 49: hyperamylasemia in 4 cases, and bleeding in 1 case). In this study, seven patients experienced stent dysfunction, which was managed via stent cleaning or the placement of a new stent.

## 5. EUS Hepaticogastrostomy

The first step of EUS-HGS is the EUS-guided puncture of a dilated intrahepatic bile duct using either a 19-gauge or 22-gauge needle. Bile is subsequently aspirated and a contrast agent is injected to confirm the accurate access. With the needle securely in place, a guidewire (either 0.025 or 0.035 inch in diameter) is introduced into the biliary tree. The tract is then dilated by either a biliary dilating balloon or a cystotome (6 Fr), in difficult situations, to ensure the successful passage of devices for biliary transumural drainage [34].

In patients with altered anatomy, EUS-HGS is a viable option for achieving biliary drainage, either as the initial choice or when standard and DAE procedures are not feasible. This was underlined in a multicenter prospective study conducted by Kitano and colleagues [22], which demonstrated that EUS-BD for MBO in patients with surgically altered anatomy (SAA) appears to be both effective and safe. It serves not only as a salvage drainage technique following unsuccessful ERCP, but also as a primary drainage method. Forty patients with SAA underwent this procedure, including those who had previously undergone gastrectomy with Roux-en-Y reconstruction (47.5%), gastrectomy with Billroth II reconstruction (15%), pancreatoduodenectomy (27.5%) and hepaticojejunostomy with Roux-en-Y reconstruction (10%).

EUS-BD was performed as the primary biliary approach in 31 patients and as a rescue biliary drainage in 9 patients. Patients treated with EUS-BD had the option of transmural stenting alone (60%), antegrade stenting alone (5%), or a combination of both techniques (35%). The technical success rate reached 100% and the clinical success rate was 95%. Patients treated with the combined procedure experienced a more favorable stent patency rate compared to those treated with a transmural stenting alone.

Another study conducted by our group [23] (Table 1) demonstrated the feasibility and safety of EUS-HGS using a newly designed partially covered self-expandable metal stent with an anti-migratory system (Figure 4). In this study, twenty-two patients were enrolled, presenting various causes of ERCP failure, including papilla infiltration by neoplastic tissue (18.2%), inaccessible papilla due to duodenal stricture (40.9%), surgically altered anatomy with Roux-en-Y reconstruction (18.2%) and incomplete biliary drainage after transpapillary stent placement (22.7%). Technical success was achieved in all patients with a mean procedural time of 43.3 ± 26.8 min. The clinical success rate was 91% (20 out of 22 patients, with a mean follow-up of 10.8 ± 3.1 months). In two patients with altered anatomy who failed to achieve a sufficient reduction in bilirubin levels, a percutaneous transhepatic biliary drainage (PTBD) approach was used to drain the right hepatic lobe. Importantly, no cases of stent misplacement or migration were observed. Nevertheless, the rate of adverse events during EUS-HGS remains high in the literature, and these events can occasionally have severe consequences, such as stent migration. Bile peritonitis is another potential complication that may occur during fistula dilation.

Sang Soo Lee et al. [35] explored the risk factor for AEs and long-term outcomes in a retrospective study enrolling 120 patients that received EUS-HGS for MBO. The adverse events evaluated were bile peritonitis, including pneumoperitoneum; hemorrhage; stent dysfunction; stent obstruction; migration; and sludges or food scraps. Concerning early adverse events, the group encountered seven episodes of stent malfunction. These included three cases of immediate stent migration and four occlusions caused by sludge or food debris, ten cases of bile peritonitis (including pneumoperitoneum) and four bleedings. Regarding late adverse events, the cohort reported 2 cases of bile peritonitis, 2 cases of a localized infected biloma and a total of 39 cases of late stent dysfunction. The most common cause of late stent dysfunction was stent obstruction (22.6%), primarily stemming from sludge or food debris, as well as migration.

## 6. EDGE (EUS-Directed Transgastric ERCP) Procedure

RYGB anatomy may be challenging when ERCP is required. An EUS-directed transgastric ERCP (EDGE) procedure consists of carrying out the standard ERCP procedure throughout a transmural fistula between the gastric pouch and the excluded stomach in order to allow for the passage of a side-view duodenoscope. The feasibility of EDGE in patients with RYGB has been reported in several reports and has been developed in the last years.

Initially, Kedia and colleagues [36] introduced a two-stage technique (referred to as double-stage EDGE). The first step involved placing a percutaneous gastrostomy (PEG) tube into the excluded stomach after identifying and distending the excluded cavity through the pouch using an EUS. Subsequently, the PEG tube was exchanged for a fully covered self-expanding metal stent (FCSEMS), and anterograde ERCP was performed via percutaneous FCSEMS. However, this method did not gain the widespread acceptance expected due to certain limitations, including the risk of PEG site infection and the inability to perform it urgently. A year later, Kedia and colleagues [37] improved their technique by developing the single-stage EDGE (SS-EDGE) thanks to the availability of a lumen-apposing metal stent (LAMS) [38]. This upgraded technique involves creating a gastro-gastric (G–G) or jejunogastric (J–G) fistula with the excluded stomach using an EUS-guided LAMS placement, eliminating the need for percutaneous access.

It is important to note that this technique does not directly interfere with the bile and pancreatic ducts. Instead, EDGE establishes a fistula by inserting a LAMS under EUS guidance between either the jejunum or the gastric pouch and the excluded stomach, followed by conventional ERCP through the LAMS using a standard duodenoscope. This technique may have potential applications in other altered anatomies by creating a fistula between the remnant stomach or the jejunum and the afferent limb.

In a multicenter study conducted by Ngamruengphong and colleagues [39] involving 23 patients, the safety and efficacy of EDGE (referred to as EUS-guided transgastric ERCP) were evaluated. The technical success rate for the placement of a 15 mm LAMS and the clinical success rate of ERCP via LAMS both reached 100%. The median wait time to perform ERCP after LAMS placement was 11 days. Stent dislodgment was observed in 33% of cases when a therapeutic duodenoscope was used, but none occurred with the slim duodenoscope. Similarly, an interim analysis by Tyberg and colleagues [40] reported a technical success rate of 100% and a clinical success rate of 91%. Unlike the previous study, they did not provide information on the waiting period between LAMS placement and ERCP, but stent dislodgment was observed in 19% of cases when a FCSEMS was used to replace the LAMS. In both studies, interventions such as over-the-scope clip (OTSC), endoscopic suturing and argon plasma coagulation (APC) were used to close the fistulous tract, while some patients were allowed to heal through secondary intention. On follow-up, there was a mean weight change ranging from −2.85 kg to −3.6 kg.

In another multicenter study by Bukhari and colleagues [41] published in 2018, the outcomes and adverse events were compared between EUS-guided gastrogastrostomy-assisted ERCP (EUS-GG-ERCP) and enteroscopy-assisted ERCP (e-ERCP) in Roux-en-Y gastric bypass (RYGB) patients. Out of 60 patients, 30 underwent EUS-GG-ERCP and the remaining 30 underwent e-ERCP (including DBE in 19 and SBE in 11). The technical success rate was higher with EUS-GG-ERCP compared to e-ERCP (100% vs. 60%, *p* < 0.001). The total procedure time and post-procedure median length of hospitalization were significantly shorter in the EUS-GG-ERCP group (49.9 min vs. 90.7 min, *p* < 0.001; and 1 day vs. 10.5 days, *p* = 0.02, respectively). However, the rate of adverse events was similar in both groups (6.7% vs. 10.0%, *p* = 1). No significant weight change was reported after EUS-GG-ERCP at a mean follow-up of 209 days.

## 7. EUS-Guided Biliary Intervention versus Enteroscopy-Assisted ERCP

Khashab and colleagues [42] carried out a retrospective study in which they compared three approaches for biliary interventions in patients with surgically altered anatomy (SAA): colonoscopy-assisted ERCP, EUS-guided biliary intervention and EUS-HGS. Their study involved 98 patients from 10 different institutions with various types of upper gastrointestinal anatomy due to prior surgeries. These included 15 patients who had undergone pancreaticoduodenectomy, 12 with Billroth II anatomy, 17 with Roux-en-Y hepaticojejunostomy, 52 with RYGB, and 2 with total gastrectomy with esophagojejunostomy. The majority of patients (70%) had Roux-en-Y anatomy, with 67% having a native papilla and 33% enteric biliary anastomosis.

In the EUS group, various interventions were performed, including EUS-HGS in 33 patients (67.4%), EUS antegrade stenting in 10 patients (20.4%), EUS-RV rendezvous technique in 2 patients (4%), EUS-HJ in 3 patients (6.1%) and hepaticoduodenostomy in 1 patient (2%). In the balloon-enteroscopy-assisted ERCP group, 5 patients (10.2%) underwent SBE, 42 (85.7%) underwent DBE and 2 (4.1%) had colonoscopy-assisted ERCP.

Clinical success was achieved in 88% of patients in the EUS group compared to 59.1% in the enteroscopy-assisted ERCP group, with an odds ratio of 2.83 (*p* = 0.03). The EUS-guided biliary intervention group also had significantly shorter procedural times (55 min vs. 95 min, *p* < 0.0001). Adverse events occurred more frequently in the EUS group (20% vs. 4%, *p* = 0.01).

These findings suggest that EUS-guided biliary interventions can be performed safely and in a more time-efficient manner. However, it is important to note that currently there are no dedicated devices for this procedure. Further extensive studies are required to provide a more comprehensive understanding of these techniques.

## 8. Conclusions

The necessity of achieving biliary drainage in patients with altered anatomy is a growing task for endoscopists given the increasing number of upper GI surgeries for bariatric interventions or malignancies.

In these situations, the endoscopic approach to bilio-pancreatic disorders is often difficult to handle and in the latest decades, several techniques have been developed in order to overcome this issue.

The first procedures described involve a forward-viewing endoscopic approach reaching initial good results with DAE-ERCP, but even with the refinement of the technique and development of new devices results remained unsatisfactory.

This need led to the development of EUS-guided interventions, especially in the last decade, with good results in terms of technical and clinical success. EUS-BD is equal to PTBD and appears to be superior of DAE-ERCP in patients with biliary obstruction and SAA.

Our aim was to focus on the need for tailoring the intervention in a case-by-case fashion, taking into account the surgery previously performed and indications considering a multidisciplinary approach with collaboration among gastroenterologists, radiologists and surgeons.

The procedure of choice should also depend on the expertise and equipment available at every institution given the long learning curve needed to perform such interventions. In this regard, the availability of EUS-dedicated tools is not uniform across different countries. However, the rapid development of the technique is rapidly increasing the diffusion of the different dedicated devices.

Moreover, the standardization of outcomes, in terms of technical and clinical success, is mandatory to make results comparable and applicable to clinical practice. The literature available is still lacking prospective comparative studies and many data come from small cohorts, while studies with abundant data, such as multicenter studies, are few.

Future research should concentrate on refining these techniques to minimize procedural complexities and enhance their efficacy. Further exploration and innovation in endoscopic approaches are vital for establishing safer and more effective strategies for biliary drainage, even in the context of altered anatomies.

## Figures and Tables

**Figure 1 diagnostics-13-03623-f001:**
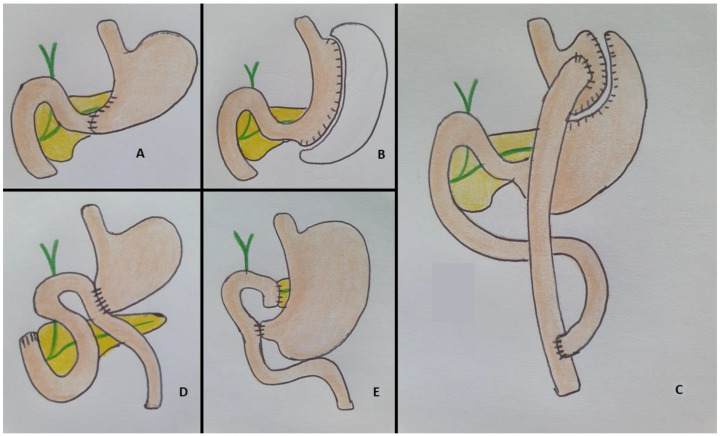
Surgically altered anatomies. Type I: (**A**) Billroth I gastrectomy; (**B**) sleeve gastrectomy. Type II: (**C**) Roux-en-Y gastric bypass; (**D**): Billroth 2 gastrectomy; (**E**): Whipple procedure.

**Figure 2 diagnostics-13-03623-f002:**
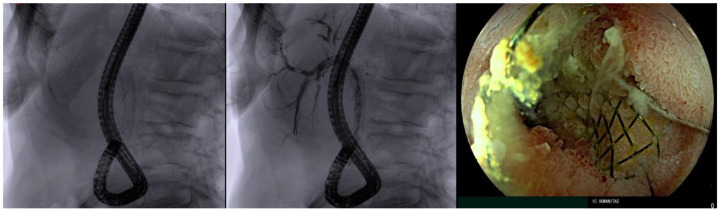
Underwater technique for reaching the papilla; radiological and endoscopic view.

**Figure 3 diagnostics-13-03623-f003:**
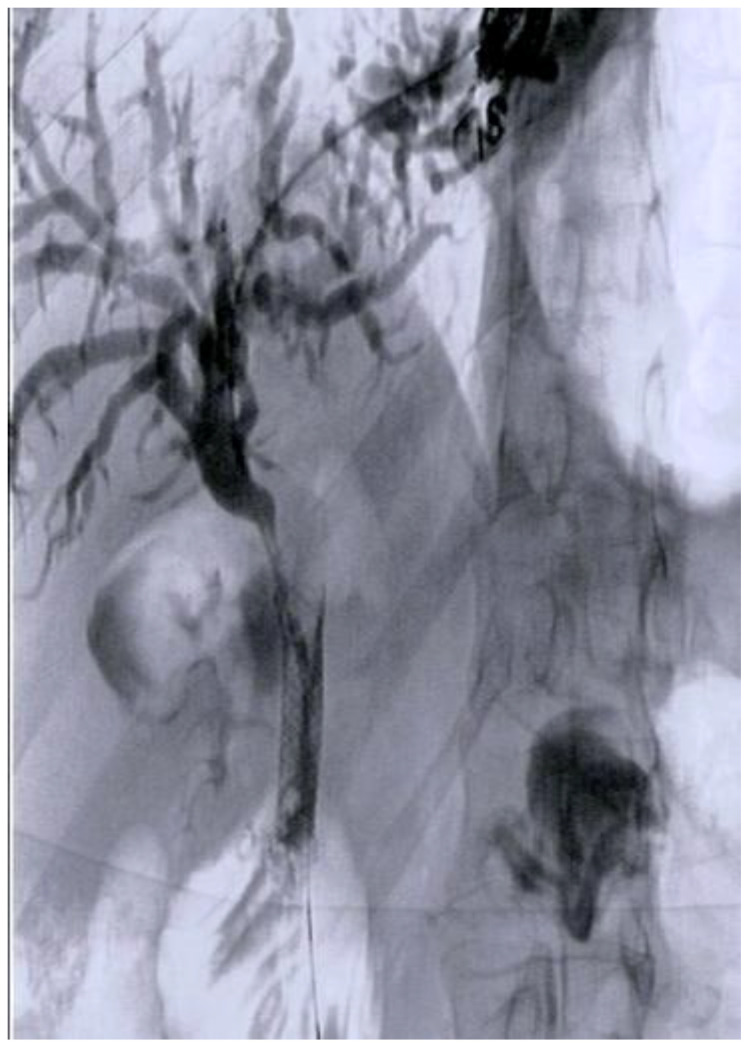
EUS-AI Biliary stenting for the treatment of unresectable malignant biliary obstruction performed at the Humanitas Research Hospital.

**Figure 4 diagnostics-13-03623-f004:**
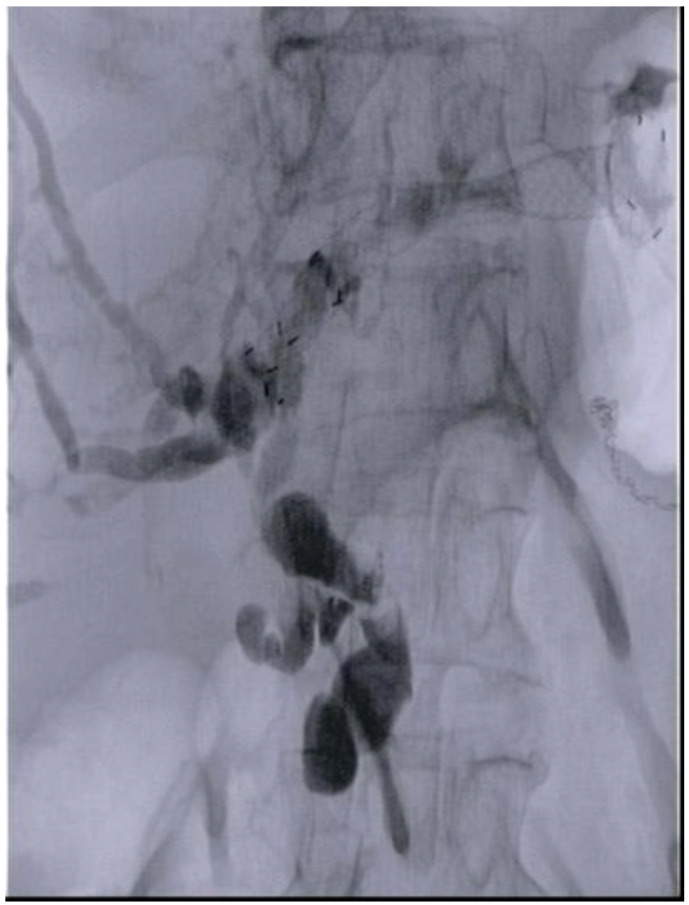
EUS-HGS with a new dedicated partially covered self-expandable metal stent with anti-migratory systems. EUS-HGS: EUS hepaticogastrostomy.

**Table 1 diagnostics-13-03623-t001:** Summary of the major topics in the literature relating biliary drainage in altered anatomy.

Authors	BD Procedure	SAA	Indication	Pts	Technical Success	Clinical Success	AEs
Simon Nennstiel [13]	**Duodenoscope** 168 (20.2%) **Pediatric colonoscope** 285 (34.3%) **SBE 144** (17.3%) **DBE** 78 (9.4%) **Colonoscope** 103 (12.4%) **Gastroscope** 50 (6%)	**RY** 186 (44%) **B II** 105 (24.8%) **Whipple 120** (28.4%)	Malignant 203 (49.4%) Benign 187 (45.5%)	441	**Duodenoscope** -B II 62 (88.1%) -Whipple 11 (57.9%) -RY 58 (75.3%) -**Pediatric Colonoscope** -B II 49 (66.2%) -Whipple 71 (64%) -Roux-en-Y 37 (37%) -**SBE** -BII - -Whipple 15 (56.6%) -RY 78 (69.6%) **DBE** -B II- -Whipple 9 (64.3%) -RY 29 (48.3%) **Colonoscope** -B II 8 (50%) -Whipple 18 (50%) -RY 28 (54.9%) **Gastroscope** -B II 33 (16.2%) -Whipple 7 (3.2%) -RY 10 (2.4%)	-	Total 4 (8%) -B II 4 (12.1%)
Fei Wang [15]	-Gastroscope -Duodenoscope -Standard colonoscope -Long-type colonoscope DBE SBE	BII 52 (53%) Subtotal or Total Gastrectomy with RY 20 (21%) Pancreatoduodenectomy or RY hepaticojejunostomy reconstruction 25 (25.8%)	Malignant 33 (34.02%) Benign 60 (61.85%)	97	**B II**—gastroscope 11/13 84.6% Duodenoscope 5/8 (62.5%) Standard colonoscope 29/31 (93.5%) **Subtotal or total gastrectomy with RY anastomosis** Standard colonoscope 2/4 (50%) Long colonoscope 7/10 (70%) DBE (5/6) 83.3% **Pancreatoduodenectomy or RY** **Hepatico-jejunostomy reconstruction** **Standard** colonoscope 3/6 (50%) Long colonoscope 88.9% (8/9) DBE 8/10 (80%)	-	Total (10/97) 10.3% 3 Pancreatitis 4 Hyperamylasemia 1 Cholangitis 1 Bleeding 1 Cardiopulmonary accident
Fugazza [17]	Pediatric colonoscope	Distal gastrectomy and RY 3/6 (50%) Whipple (Pylorus preserving) 2/6 (33.3%) Gastrojejunal Bypass 1/6 (16.7%)	Benign 6/6	6	100%	100%	0
Takaaki Fujimoto [18]	DBE Gastroscope	Gastrectomy and RY 38 (37.2%); BII 24 (23.5%); Pancretoduodenoctomy followed by BII 23 (22.5%); Pancretoduodenoctomy or RY hepaticojejunostomy 17 (16.6%)	Benign 100%	102	88% (144/164)	\	11/180 (6%) 2 Perforation 7 Cholangitis 2 Hyperamylasemia
Zouhairi [19]	RA-ERCP	33 RY (91.7%) 2 B II (5.5%) 1 Hepaticojejunostomy (2.93%)	Bengin 100%	36	29/32 (89.7%)	\	10/42 (23.8%) 3 Nausea and abdominal pain 7 Pancreatitis
Iwashita [20]	EUS-AI	14 Gastrectomy with RY 1 Gastrectomy with BII 1 Hepatectomy with biliary reconstruction 4 Gastric bypass	Malignant 100%	20	19/20 95%	19/20 95%	4/20 (20%) 3 Mild pancreatitis 1 Mild fever
Iwashita [21]	EUS AI PTBD	Gastrectomy with RY 49 (76.6%) Gastrectomy with BII 8 (11.8%) Gastric bypass 7 (10.3%)	Malignant 100%	64	**EUS AI** 34/35 (97.1%) **PTBD** (28/29) 96.6%	EUS AI 34/35 (97.1%) PTBD (27/29) 93.1%	EUS AI 4/34 (11.4%) PTBD 8/29 (27.6%)
Minaga [22]	EUS AI EUS HGS Combination technique	Gastrectomy with RY 19/40 (47.5%) Gastrectomy with BII 6/40 (15%) Pancreaticoduodenectomy 11/40/27.5%) Hepaticojejunostomy with RY 4/40 (10%)	Malignant 100%	40	**EUS HGS** 24 60% **EUS AI** 2 5% **Combination technique** 14 35%	38 [95,(83.1–99.4)]	Early AEs 6 (15%) 3 Bile leak 1 Bile peritonitis 2 Pneumoperitoneum Late AEs 6 (15%) 1 Jejunal ulcer 5 Stent disfunction
Anderloni [23]	EUS HGS	4 RY	Malignant 100%	22	100%	20/22 (91%)	3/22 (13.6) Hepatic abscess

BD: biliary drainage; SAA: surgically altered anatomy; AEs: adverse events; Pts: patients; SBE: single-balloon enteroscopy; DBE: double-balloon enteroscopy; RA-ERCP: rotational assisted enteroscopy device; RY: Roux-en-Y; RYGB: Roux-en-Y gastric bypass; EUS AI: EUS-guided antegrade intervention; BII Billroth II; PTBD: percutaneous transhepatic biliary drainage; EUS HGS: EUS hepaticogastrostomy.

## Data Availability

Not applicable.

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
