# Peer review of "Endoscopic Biliary Drainage in Surgically Altered Anatomy"

_diagnostics, 2023, doi:10.3390/diagnostics13243623_

Round 1

Reviewer 1 Report

Comments and Suggestions for Authors

1.      This manuscript provides a valuable overview of the challenges and evolving techniques in the field of biliary drainage for patients with surgically altered anatomy. It emphasizes the importance of tailoring interventions on a case-by-case basis, taking into account surgical history and available resources.

2.      The manuscript mentions success rates and complications associated with various techniques but does not provide specific data on these outcomes. Quantitative data would be beneficial for a more in-depth analysis.

3.      While the article discusses the need for specialized devices and equipment for EUS-guided procedures, it doesn't evaluate further into the availability of these resources in different healthcare settings.

Comments on the Quality of English Language

Most are fine, minor mis-spelling  may need some editing.

Author Response

Thank you very much for taking the time to review this manuscript. We appreciate the feedback from the reviewer. Concerning the first point, we have included a table detailing the adverse events reported in the various studies. Regarding the second point, considering this is an evolving scenario (in particularly for EUS tools) we couldn’t report reliable info about equipment availability. However, we better discussed this important point

Reviewer 2 Report

Comments and Suggestions for Authors

I would like you to modify the abstract which is like an introduction and does not say anything about the new techniques you describe. Does Figure 3 belong to you? It is understood from the text that it is Ishiwata's and you should have his permission. I would like a table with the studies on the endoscopic approach after Roux-y anastomoses. Conclusions should be clearer and contain future directions. I want to congratulate you for the article that shows a rich experience in modern invasive endoscopy techniques.

Author Response

Thank you very much for taking the time to review this manuscript. We appreciate the feedback and have made the necessary modifications. The abstract has been revised to emphasize the use of EUS and various techniques. Additionally, we've incorporated a table and adjusted the conclusions as suggested. Concerning the figure, we have clarified that it came from our hospital.

Reviewer 3 Report

Comments and Suggestions for Authors

The patient with surgically altered anatomy is a challenging case for experimented endoscopists. The present review accurately describes the limited endoscopic options for performing ERCP for these patients. The new EUS-assisted techniques along with more usable devices are a good help for these patients.

The paper is well conducted with appropriate references.

Author Response

Thank you very much for taking the time to review this manuscript, and thanks for your support
